# Chymase as a Novel Therapeutic Target in Acute Pancreatitis

**DOI:** 10.3390/ijms222212313

**Published:** 2021-11-15

**Authors:** Toru Kuramoto, Denan Jin, Koji Komeda, Kohei Taniguchi, Fumitoshi Hirokawa, Shinji Takai, Kazuhisa Uchiyama

**Affiliations:** 1Department of General and Gastroenterological Surgery, Osaka Medical and Pharmaceutical University, Takatsuki-City 569-8686, Japan; toru.kuramoto@ompu.ac.jp (T.K.); koji.komeda@ompu.ac.jp (K.K.); fumitoshi.hirokawa@ompu.ac.jp (F.H.); uchi@ompu.ac.jp (K.U.); 2Department of Innovative Medicine, Osaka Medical and Pharmaceutical University, Takatsuki-City 569-8686, Japan; denan.jin@ompu.ac.jp; 3Translational Research Program, Osaka Medical and Pharmaceutical University, Takatsuki-City 569-8686, Japan; kohei.taniguchi@ompu.ac.jp

**Keywords:** acute pancreatitis, chymase, inhibitor, inflammation, survival rate, hamster

## Abstract

Acute pancreatitis is still a life-threatening disease without an evidenced therapeutic agent. In this study, the effect of chymase in acute pancreatitis and the possible effect of a chymase inhibitor in acute pancreatitis were investigated. Hamsters were subcutaneously administered 3.0 g/kg of L-arginine to induce acute pancreatitis. Biological markers were measured 1, 2, and 8 h after L-arginine administration. To investigate the effect of a chymase inhibitor, a placebo (saline) or a chymase inhibitor TY-51469 (30 mg/kg) was given 1 h after L-arginine administration. The survival rates were evaluated for 24 h after L-arginine administration. Significant increases in serum lipase levels and pancreatic neutrophil numbers were observed at 1 and 2 h after L-arginine administration, respectively. Significant increases in pancreatic neutrophil numbers were observed in the placebo-treated group, but they were significantly reduced in the TY-51469-treated group. A significant increase in the pancreatic tumor necrosis factor-α mRNA level was observed in the placebo-treated group, but it disappeared in the TY-51469-treated group. Chymase activity significantly increased in the placebo-treated group, but it was significantly reduced by treatment with TY-51469. The survival rate significantly improved in the TY-51469-treated group. A chymase inhibitor may become a novel therapeutic agent for acute pancreatitis.

## 1. Introduction

Acute pancreatitis is a systemic inflammatory disease triggered by autolysis due to the activation of pancreatic enzymes. Although the mortality of acute pancreatitis has become lower than 5% with the development of international guidelines for the management of acute pancreatitis, the mortality rate of severe acute pancreatitis is still 10–20% [1,2,3,4]. Most guidelines for acute pancreatitis only mention supportive therapies such as fluid management and nutritional intervention, and they do not mention therapeutic agents [1,3,4]. The Japanese guideline for treating acute pancreatitis mentioned a trypsin-like protease inhibitor, gabexate mesylate, as a possible therapeutic agent, but no significant reduction in mortality was reported in a meta-analysis of clinical studies [2]. Therefore, there is no reliable agent that can be recommended at present.

Several agents have been studied to prevent acute pancreatitis using animal models of the condition [5,6,7]. In a rat taurodeoxycholate-induced acute pancreatitis model, activated mast cells and increased mRNA levels of tumor necrosis factor (TNF)-α were observed in the pancreas, but they were prevented by pretreatment with a mast cell stabilizer, cromolyn [5]. In a rat cerulein-induced acute pancreatitis model, mast cell numbers significantly increased in the pancreas [6]. Treatment with a mast cell stabilizer, ketotifen, before an injection of cerulein, significantly reduced neutrophil infiltration into the pancreas in the model, along with a significant attenuation of the mast cell number [6]. These results indicate the significance of mast cells in initiating acute pancreatitis. However, mast cells include several inflammatory mediators, such as histamine, serotonin, cytokines, and serine proteases. It remains unclear which factor might play an important role in the initiation of acute pancreatitis.

Chymase is a chymotrypsin-like serine protease that is stored in mast cell granules in normal states. Stimuli of inflammation and injury activate mast cells, and then chymase is released from the granules into the surrounding tissues along with other inflammatory mediators [8]. Originally, chymase was discovered as an angiotensin II-generating enzyme other than an angiotensin-converting enzyme (ACE) in human arteries [9]. However, chymase has also been shown to activate pro-matrix metalloprotease (proMMP-9) to matrix metalloproteinase (MMP)-9, which cleaves the extracellular matrix and induces neutrophil infiltration [10,11]. In acute pancreatitis models, MMP-9 was significantly increased in the pancreas, and the severity of pancreatitis was significantly reduced by treatment with an MMP inhibitor and in MMP-9-deficient mice [12,13]. Previous reports have demonstrated that chymase inhibition significantly reduced MMP-9 levels and attenuated the severity of acute inflammatory diseases, such as indomethacin-induced small intestinal inflammation and acute liver failure [10,11]. Therefore, chymase may become a valuable target for the amelioration of acute pancreatitis. 

In the present study, several inflammatory factors, such as serum lipase, TNF-α, and neutrophil infiltration, were measured after injection of L-arginine to study the dose of L-arginine and the timing of administration of the chymase inhibitor, and then a specific chymase inhibitor, TY-51469, was evaluated using this acute pancreatitis model.

## 2. Results

### 2.1. Two Doses of L-Arginine-Induced Acute Pancreatitis Models

Two doses of L-arginine-induced acute pancreatitis models were compared. Low and high-dose groups were intraperitoneally injected with 1.5 and 3.0 g/kg, respectively. Serum lipase levels and mRNA levels of the inflammatory mediator TNF-α were examined 12 h after L-arginine injection to determine the dose in subsequent studies. Each group started with six hamsters, but two hamsters died within 12 h in the high-dose group (Figure 1A).

Both serum lipase and TNF-α mRNA levels were significantly higher in the high-dose group than in the low-dose and control groups (Figure 1B). However, there was no significant difference between the low-dose group and the control group.

### 2.2. Time Course of the High-Dose L-Arginine-Induced Model of Acute Pancreatitis

Serum lipase and TNF-α mRNA levels were measured before and 1, 2, and 8 h after L-arginine injection. At each time point, the group started with six hamsters, but one hamster died between 2 and 8 h.

Serum lipase was significantly increased even at 1 h after L-arginine injection, increasing continuously after that (Figure 2A). The TNF-α mRNA level in the pancreas was also significantly increased 1 h after L-arginine injection, and significant increases were observed at 2 and 8 h (Figure 2B).

Representative images of the pancreas immunostained with anti-neutrophil elastase antibody (neutrophil) before and 8 h after the injection of L-arginine are shown in Figure 2C. The number of neutrophils in the tissue sections tended to be higher 1 h after L-arginine injection, and significant increases were observed at 2 and 8 h (Figure 2D).

### 2.3. Effects of Chymase Inhibitor on Serum Lipase and Pancreatic TNF-α mRNA Levels

Serum lipase was significantly higher in the placebo-treated group than in the control group, but there was no significant difference between the placebo- and TY-51469-treated groups (Figure 3A). A significant increase in the pancreatic TNF-α mRNA level was observed in the placebo-treated group compared to the control group, but it tended to be attenuated in the TY-51469-treated group (Figure 3B).

### 2.4. Chymase Inhibitor TY-51469 Decreases Immune Cell Infiltration into the Pancreas

Representative images in the pancreas stained with HE and anti-neutrophil elastase-positive cells (neutrophils) in the control, placebo-, and TY-51469-treated hamsters are shown in Figure 4A. Numbers of neutrophils in the tissue sections were significantly higher in the placebo-treated group than in the control group, but they were significantly lower in the TY-51469-treated group than in the placebo-treated group (Figure 4B).

### 2.5. Chymase Inhibitor TY-51469 Suppressed Pancreatic Chymase and MMP-9 Activities

Chymase activity in the pancreas was significantly increased in the placebo-treated group compared to the control group, but it was significantly reduced by treatment with TY-51469 (Figure 5A). MMP-9 activity tended to be higher in the placebo-treated group than in the control group, but it tended to be lower in the TY-51469-treated group (Figure 5B). 

### 2.6. Chymase Inhibitor TY-51469 Decreases the Number of Chymase-Positive Cells in the Pancreas

Representative images of tissue sections stained with toluidine blue (mast cells) and immunostained with anti-chymase antibody (chymase-positive cells) from the control, placebo-, and TY-51469-treated hamsters are shown in Figure 6A. Numbers of chymase-positive cells in the tissue sections were significantly lower in the TY-51469-treated group than in the placebo-treated group (Figure 6B).

### 2.7. Chymase Inhibitor TY-51469 Increases the Survival Rate of Hamsters

The survival rates of the placebo- and TY-51469-treated groups were 26% and 56% 24 h after the injection of L-arginine, respectively, and the difference was significant (Figure 7).

## 3. Discussion

This study aimed to evaluate the effects of a chymase inhibitor on acute pancreatitis and the survival rate. Acute pancreatitis models in various rodents have been reported [14,15,16]. In a study using hamsters, sodium taurocholate-, alcohol + palmitoleic acid-, and cerulein-induced acute pancreatitis models showed similar mild pancreatitis, and L-arginine is a model for severe necrotizing pancreatitis [17]. In the present study, a hamster L-arginine-induced acute pancreatitis model, which has been confirmed to be fatal after the onset of pancreatitis in some animals, was selected to evaluate the effect of a chymase inhibitor on the survival rate [17]. Among the acute pancreatitis models using various rodents, a hamster model was used because of the species differences in chymase [18,19]. For example, human and hamster chymases convert angiotensin I to angiotensin II, but rat chymase does not [18,19]. Enzymatic characterization of hamster chymase among rodents shows that it resembles human chymase the most [18]. For example, the conversion of angiotensin I to angiotensin II in an extract from rat arteries was completely inhibited by an ACE inhibitor [19]. However, the conversion in an extract from humans and hamsters was partially inhibited by an ACE inhibitor. It was entirely blocked by the combination of an ACE inhibitor and a chymase inhibitor [19]. In a rat L-arginine-induced acute pancreatitis model, pretreatment with an ACE inhibitor attenuated acute pancreatitis, reduced TNF-α, and infiltrated inflammatory cells in the pancreas [20]. Angiotensin II induces oxidative stress via nicotinamide adenine dinucleotide phosphate (NADPH) oxidase in the pancreas and may be involved in the pathogenesis of acute pancreatitis [21]. Therefore, a hamster L-arginine-induced acute pancreatitis model was considered appropriate for use in this study.

Treatment was administered 1 h after the onset of pancreatitis in this study, although most reports of treatment agents for acute pancreatitis were administered before the start [14,15,16,20]. In clinical practice, patients usually visit the hospital after the onset of acute pancreatitis, reporting abdominal pain. At that time, increased markers of pancreatitis are observed, and treatment is then started. Serum lipase secreted by the acinar cells of the pancreas has been widely used to establish the diagnosis of acute pancreatitis [22]. From this perspective, it was necessary to investigate the efficacy of the chymase inhibitor after the onset of acute pancreatitis had been confirmed. In the present study, serum lipase was significantly increased 1 h after the injection of 3.0 g/kg of L-arginine, and it was further increased at 2 h, as shown in Figure 2. A significant increase in the pancreatic TNF-α mRNA level was also observed even at 1 h after L-arginine injection. Neutrophil infiltration into the pancreas tended to be increased at 1 h, but there was no significant difference compared to that before L-arginine injection. A significant increase in neutrophil infiltration was observed after 2 h. Based on these findings, the decision was made to administer the chymase inhibitor 1 h after L-arginine injection.

TY-51469 is a specific chymase inhibitor that inhibits human chymase with an IC_50_ of 7 nM, but it does not impede other chymotrypsin-like serine proteases, bovine chymotrypsin, and cathepsin G, even at a concentration of 10 μM [21]. TY-51469 has been reported to improve inflammatory diseases, such as indomethacin-induced small intestinal inflammation and acute liver failure, along with the reduction of MMP-9 [9,10]. Chymase can directly activate the inactive precursor form, proMMP-9, to the active state, MMP-9 [23]. Neutrophil secretory granules contain proMMP-9, which is released from the granules after activating neutrophils and then the proteolytic conversion of proMMP-9 to the active form [24]. MMP-9 is involved in the degradation and remodeling of the extracellular matrix and the pathogenesis of acute pancreatitis [12,13]. Previous reports have demonstrated that MMP-9 was significantly increased in the pancreas in an acute pancreatitis model, and both the severity of pancreatitis and the infiltration of leukocytes into the pancreas were reduced considerably in treatment with an MMP inhibitor and MMP-9-deficient mice [12,13]. In the present study, pancreatic MMP-9 activity was increased after L-arginine injection, but its activity was attenuated by treatment with TY-51469, although the difference was not significant, as shown in Figure 5. Furthermore, the significant increase in neutrophil infiltration into the pancreas observed in the placebo-treated acute pancreatitis model hamsters was significantly ameliorated by treatment with TY-51469, as shown in Figure 4. The mechanism by which the chymase inhibitor decreased the neutrophil infiltration of acute pancreatitis might depend on inhibiting MMP-9 activation in the pancreas.

Chymase is stored as an inactive enzyme in secretory granules of mast cells, where the pH within the granule is regulated at pH 5.5 [25]. The optimal pH for chymase activity is between 7 and 9. Its maximum level of enzymatic activity occurs immediately upon release from the granules when the mast cells are activated by inflammation [18]. A significant increase in mast cell numbers was observed in the placebo-treated acute pancreatitis model in the present study. A similar finding was reported in a rat cerulein-induced acute pancreatitis model. The severity of pancreatitis and mast cell numbers were significantly reduced by pretreatment with a mast cell stabilizer [6]. There was also a significant reduction in mast cell numbers in the TY-51469-treated hamsters in the present study. Several enzymatic functions of chymase have been demonstrated, and chymase may play a crucial role in increasing mast cell numbers [26].

In addition to the formation of angiotensin II and MMP-9, chymase also forms stem cell factor (SCF) by proteolytically cleaving the inactive membrane-bound precursor form of SCF to the active form [26]. SCF induces the development and proliferation of immature mast cells to mature mast cells [26]. An increase in chymase activity may increase chymase-positive mast cell numbers, and, conversely, the inhibition of chymase may result in reduced chymase-positive mast cell numbers. Therefore, the reduction of chymase-positive mast cell numbers by treatment with TY-51469 may also be involved in the mechanism of ameliorating acute pancreatitis.

The survival rate after the injection of L-arginine was significantly improved by treatment with TY-51469, as shown in Figure 7. The development of systemic complications in acute pancreatitis is mainly responsible for the survival rate associated with this disease. The systemic damage encountered in acute pancreatitis is similar to that occurring in patients with multiple organ failure thought to be related to the overproduction of proinflammatory cytokines. In particular, the proinflammatory cytokine, TNF-α, may play a central role in acute pancreatitis and mediate the systemic damage caused by the disease [27]. Increased TNF-α levels were observed in a rat acute pancreatitis model, and improvement in the survival rate was shown by TNF-α blockade using an anti-TNF-α neutral antibody [27]. TNF-α may be linked to the mortality of acute pancreatitis. In the present study, a significant increase in TNF-α levels was observed in the placebo-treated acute pancreatitis model hamsters, but it was ameliorated by treatment with the chymase inhibitor. On the other hand, an early increase in MMP activity in the peritoneal fluid that coincided with increased lung permeability and distant organ injury following acute pancreatitis was observed, but they were significantly attenuated by the MMP inhibitor, which increased the survival rate [28]. In addition, a clinical report implicated MMP-9 in increased end-organ injury during acute pancreatitis [29]. On the other hand, acute pancreatitis causes inflammation not only in the pancreas but also in other organs, such as the lungs, and it is possible that the chymase inhibitors suppressed inflammation in the other organs as well. In fact, chymase inhibitors, including TY-51469, have been shown to attenuate several inflammatory diseases other than pancreatitis [30]. Chymase inhibition may improve the survival rate through a variety of mechanisms in acute pancreatitis. 

## 4. Material and Methods

### 4.1. Drug

TY-51469 was synthesized as a specific chymase inhibitor and obtained from Toaeiyo Ltd. (Tokyo, Japan) [21]. L-arginine was purchased from FUJIFILM Wako Pure Chemical Co. (Osaka, Japan).

### 4.2. Animals and Experimental Design

Six-week-old male hamsters weighing 90–110 g were obtained from Japan SLC (Shizuoka, Japan). The hamsters were fed regular hamster chow and tap water ad libitum, and they were housed in a temperature-, humidity-, and light-controlled room. To establish a hamster acute pancreatitis model, L-arginine (1.5 g/kg or 3.0 g/kg) was injected intraperitoneally twice hourly.

To evaluate the effect of TY-51469 on acute pancreatitis, 30 mg/kg of TY-51469 dissolved in saline or placebo (saline) was administered 1 h after the second injection of L-arginine, and blood and pancreas samples were evaluated 8 h after the injection of L-arginine.

To determine the effect of TY-51469 on the survival rate of L-arginine-induced pancreatitis, 30 mg/kg TY-51469 or placebo was administered 1 h after the second injection of L-arginine, and the resultant survival rates were calculated by dividing the number of hamsters that survived every 4 h after L-arginine administration by the number of hamsters before the start of the study.

All animal procedures were approved by the Committee of Animal Use and Care of Osaka Medical and Pharmaceutical University (Approval number: 21030-A, 20 April 2021) and performed in accordance with the institution’s Guidelines for Animal Research.

### 4.3. Serum Lipase

Serum was separated from the blood samples, and the serum lipase level was measured by SRL Co., Ltd. (Tokyo, Japan).

### 4.4. Preparation of the Pancreas

After anesthetizing the animals with sodium pentobarbital (50 mg/kg, i.p.), the pancreas was removed. The tissue was fixed with Carnoy’s fixative for histochemical analysis. The remaining portion of the pancreas was used for the measurement of mRNA levels and enzyme activity.

### 4.5. Histological Analysis

The fixed tissue was paraffin-embedded, and sections were prepared at a thickness of 3 µm. The sections were stained with hematoxylin-eosin (HE) and toluidine blue to identify mast cells [31,32]. To measure the numbers of neutrophils and chymase-positive cells, immunohistochemical staining was performed using an anti-chymase antibody (Otsuka Pharmaceutical Co., Ltd., Tokushima, Japan) and an anti-neutrophil elastase antibody (Santa Cruz Biotechnology, Inc., Dallas, TX, USA) [31,32]. Sections were incubated for 1 h at room temperature with anti-chymase antibody or anti-neutrophil elastase antibody C, followed by reaction with components from a labeled streptavidin-biotin peroxidase kit, including 3-amino-9-ethylcarbazole color development (Dako LSAB kit, DAKO, Carpinteria, CA, USA). The sections were lightly counterstained with hematoxylin. The number of neutrophil elastase-positive cells (neutrophils) or chymase-positive cells within the pancreas was quantified using a computerized morphometry system.

### 4.6. Real-Time Reverse Transcription Polymerase Chain Reaction (RT-PCR) 

Total RNA was extracted from the pancreas using the Trizol reagent (Life Technologies, Rockville, MD, USA) and subsequently dissolved in RNase-free water (Takara Bio Inc., Otsu, Japan). The total RNA (2.5 μg) was transcribed into cDNA with Superscript VIRO (Invitrogen, Carlsbad, CA, USA). The mRNA level was measured by real-time RT-PCR on a Stratagene Mx3000P (Agilent Technologies, San Francisco, CA, USA) using TaqMan fluorogenic probes. Primers and probes for real-time RT-PCR of TNF-α and 18S ribosomal RNA (rRNA) were designed by Roche Diagnostics (Tokyo, Japan) [33]. The primers were as follows: 5′-ctgagccatcgtgccaat-3′ (forward) and 5′-ccagctggttgtctttgaga-3′ (reverse) for TNF-α and 5′-aactgcagcaactttaatatacgc-3′ (reverse) for 18S rRNA. The probes were as follows: 5′-cctcctgg-3′ for TNF-α and 5′-cagcagcc-3′ for 18S rRNA. The mRNA level of TNF-α was normalized to that of 18S rRNA.

### 4.7. Enzyme Activities

The pancreatic tissues were homogenized in 20 mM Na-phosphate buffer, pH 7.4 [34], and then the homogenate was centrifuged at 8000× *g* for 30 min. The supernatant was discarded, and the pellet was resuspended and homogenized in 10 mM Na-phosphate buffer, pH 7.4, containing 2 M KCl and 0.1% Nonidet P-40. The homogenate was centrifuged at 8000× *g* for 30 min, and the supernatant was used for enzyme activity measurements.

Chymase activity was measured according to a previously described method [34]. The pancreatic extract was incubated with angiotensin I at 37 °C for 60 min, and then trichloroacetic acid was added to stop the reaction. One unit of chymase activity was defined as the amount of enzyme required to form 1 μmol of angiotensin II/min. MMP-9 activity was measured using an MMP-9 activity assay kit (QuickZyme Biosciences, Leiden, The Netherlands).

The protein concentration in the pancreatic extract was measured by BCA Protein Assay Reagents (Pierce, Rockford, IL, USA), using bovine serum albumin as a standard.

### 4.8. Statistical Analysis

The data are expressed as the means ± standard error of the mean (SEM) and they were analyzed with bell curve statistical analysis software for Microsoft Excel (Tokyo, Japan). Significant differences among mean values of multiple groups were evaluated using a one-way analysis of variance followed by Fisher’s protected least significant difference (LSD) test. The survival rate was evaluated using the Kaplan–Meier method. Values of *p* < 0.05 were considered significant.

## 5. Conclusions

The administration of a chymase inhibitor improved the survival rate by suppressing inflammatory mediators and neutrophil infiltration into the pancreas in a hamster model of L-arginine-induced acute pancreatitis. This study demonstrates that chymase inhibition may become a novel therapeutic strategy for the treatment of acute pancreatitis.

## Figures and Tables

**Figure 1 ijms-22-12313-f001:**
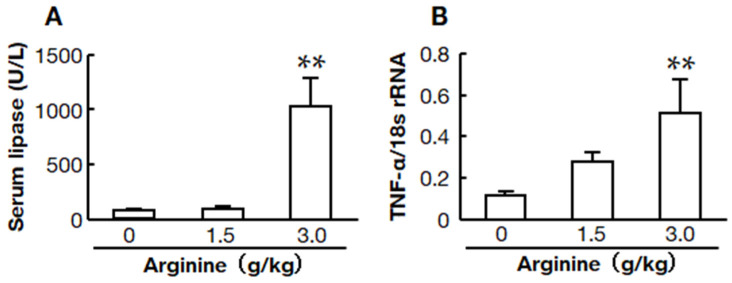
Dose responses of L-arginine on serum lipase and pancreatic TNF-α mRNA levels. Serum lipase (**A**) and pancreatic TNF-α mRNA levels (**B**) in hamsters 12 h after injections of 0, 1.5, and 3.0 mg/kg L-arginine (0 mg/kg: *n* = 6, 1.5 mg/kg: *n* = 6, 3.0 mg/kg: *n* = 4). Values represent means ± SEM. ** *p* < 0.01 vs. 0 mg/kg of L-arginine.

**Figure 2 ijms-22-12313-f002:**
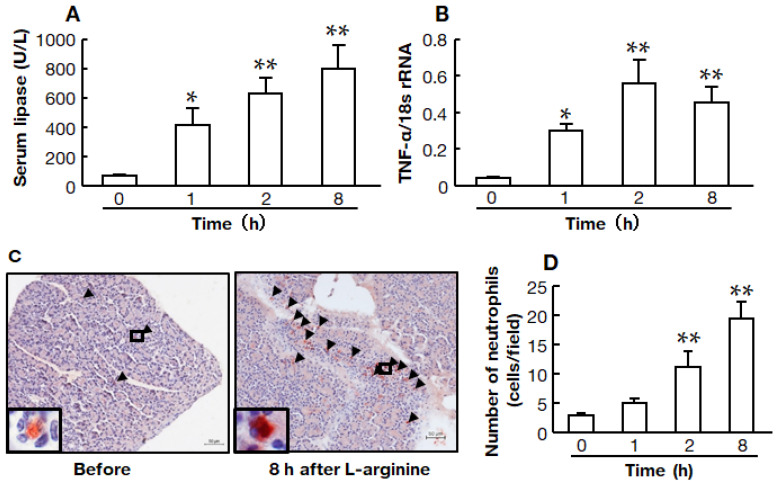
Time courses of serum lipase, pancreatic TNF-α mRNA levels, and neutrophil numbers after L-arginine injection. Serum lipase (**A**) and pancreatic TNF-α mRNA levels (**B**) in hamsters before and 1, 2, and 8 h after the injection of 3.0 mg/kg L-arginine (before: *n* = 6, 1 h: *n* = 6, 2 h: *n* = 6, 8 h *n* = 5). Representative images of a pancreas immunostained with anti-neutrophil elastase antibody from hamsters before (left image) and 8 h (right image) after the injection of L-arginine (**C**). Arrows indicate neutrophil elastase-positive cells. Original magnification 200× (inset: 1000×) (**C**). Number of neutrophil elastase-positive cells (neutrophils) in the pancreas before and 1, 2, and 8 h after the injection of 3.0 mg/kg L-arginine (before: *n* = 6, 1 h: *n* = 6, 2 h: *n* = 6, 8 h: *n* = 5) (**D**). Values represent means ± SEM. * *p* < 0.05 and ** *p* < 0.01 vs. before.

**Figure 3 ijms-22-12313-f003:**
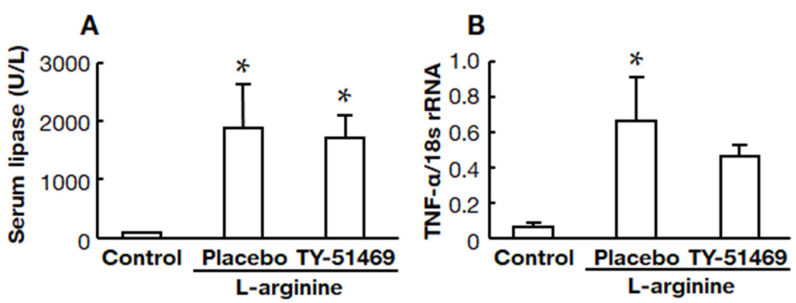
Effects of TY-51469 on serum lipase and pancreatic TNF-α mRNA levels. Serum lipase (**A**) and pancreatic TNF-α mRNA levels (**B**) in control, placebo-, and TY-51469-treated groups 8 h after the injection of L-arginine (control group: *n* = 6, placebo-treated group: *n* = 8, TY-51469-treated group: *n* = 8). Values represent means ± SEM. * *p* < 0.05 vs. control group.

**Figure 4 ijms-22-12313-f004:**
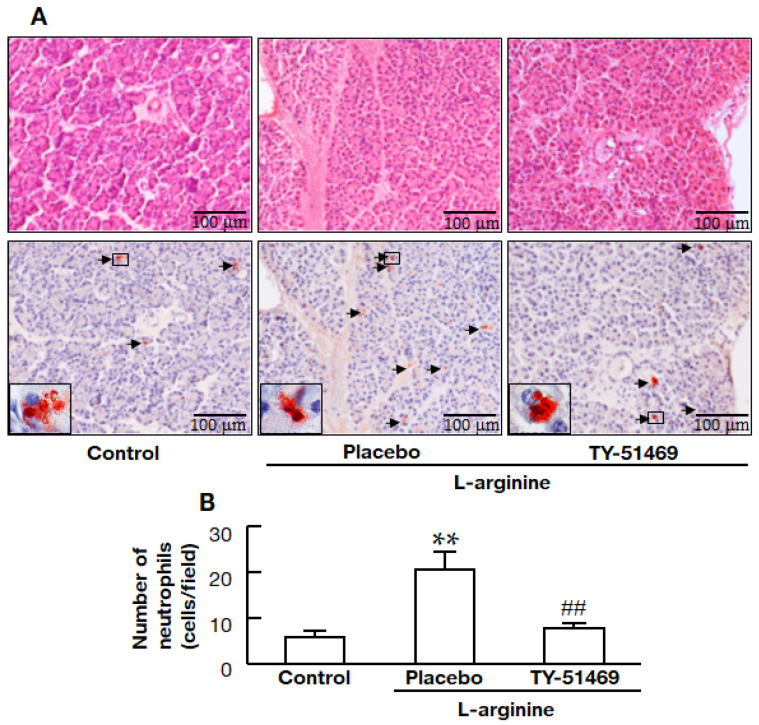
Effects of TY-51469 on neutrophil numbers in the pancreas. Representative images of a pancreas stained with HE stain (upper images) and immunostained with anti-neutrophil elastase antibody (lower images) from the control, placebo-, and TY-51469-treated hamsters 8 h after the injection of L-arginine (**A**). Arrows indicate neutrophil elastase-positive cells (**A**: lower images). Original magnification 200× (inset: 1000×) (**A**). The number of neutrophil elastase-positive cells (neutrophils) in the pancreas in control, placebo-, and TY-51469-treated groups 8 h after the injection of L-arginine (control group: *n* = 6, placebo-treated group: *n* = 8, TY-51469-treated group: *n* = 8) (**B**). Values represent means ± SEM. ** *p* < 0.01 vs. control group. ## *p* < 0.01 vs. placebo-treated group.

**Figure 5 ijms-22-12313-f005:**
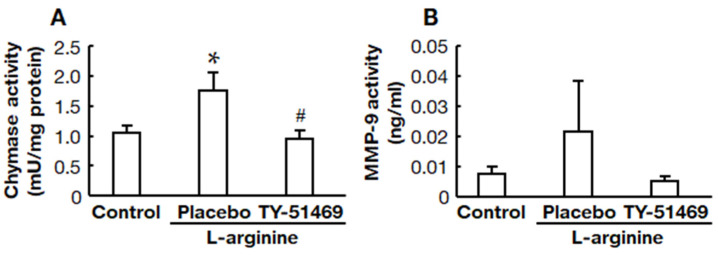
Effects of TY-51469 on chymase and MMP-9 activities in the pancreas. Pancreatic chymase (**A**) and MMP-9 (**B**) activities in the control, placebo-, and TY-51469-treated groups 8 h after the injection of L-arginine (control group: *n* = 6, placebo-treated group: *n* = 8, TY-51469-treated group: *n* = 8). Values represent means ± SEM. * *p* < 0.05 vs. control group. # *p* < 0.05 vs. placebo-treated group.

**Figure 6 ijms-22-12313-f006:**
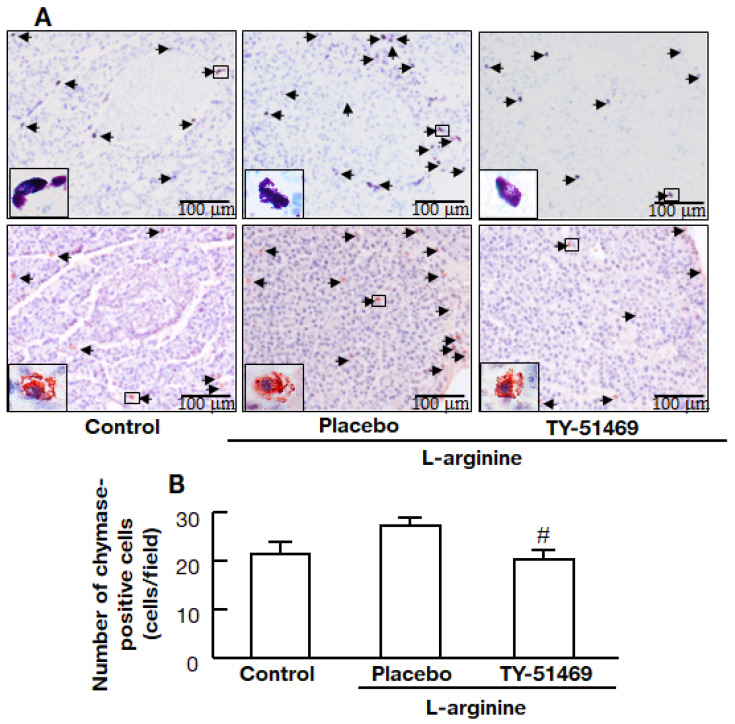
Effects of TY-51469 on numbers of chymase-positive cells. Representative images of pancreata stained with toluidine blue (upper images) and immunostained with anti-chymase antibody (lower images) from the control, placebo-, and TY-51469-treated hamsters 8 h after the injection of L-arginine (**A**). The arrows show toluidine blue-stained cells (mast cells) and chymase-positive cells (**A**). Original magnification 200× (inset: 1000×) (**A**). The number of chymase-positive cells in control, placebo-, and TY-51469-treated groups 8 h after the injection of L-arginine (control group: *n* = 6, placebo-treated group: *n* = 8, TY-51469-treated group: *n* = 8) (**B**). Values represent means ± SEM. # *p* < 0.05 vs. placebo-treated group.

**Figure 7 ijms-22-12313-f007:**
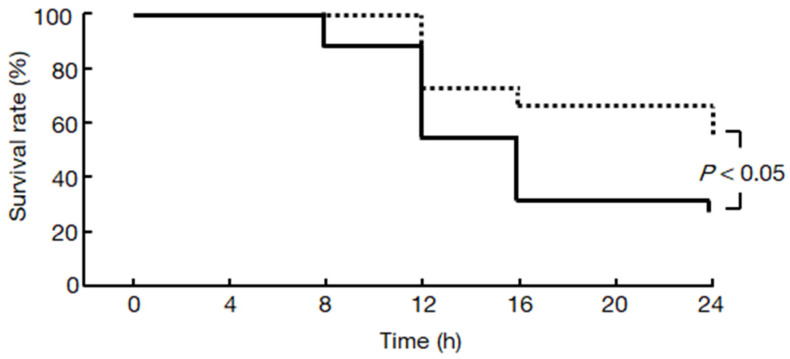
Effect of TY-51469 on the survival rate. Cumulative percent survival rate in the placebo (solid line)- and TY-51469 (dotted)-treated groups after the injection of L-arginine (placebo-treated group: *n* = 25, TY-51469-treated group, *n* = 25). The difference between the placebo- and TY-51469-treated groups is significant (Kaplan–Meier analysis followed by the log-rank test, *p* < 0.05).

## Data Availability

The data that support the findings of this study are available upon request from the corresponding author, Shinji Takai (E-mail: shinji.takai@ompu.ac.jp).

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
