# Peer review of "Chymase as a Novel Therapeutic Target in Acute Pancreatitis"

_ijms, 2021, doi:10.3390/ijms222212313_

Round 1

Reviewer 1 Report

Kuramoto et al. found increased Chymase activity upon Arginine (pancreatitis-inducing agent) administration, that was significantly reduced by treatment with a chymase inhibitor TY-51469. The survival rate was significantly improved in the TY-51469-treated group. The authors propose TY-51469 as a potential novel therapeutic to treat pancreatitis.

Before being further considered for publication, the authors should address the following comments:

  1. Abstract- please correct "the affect" to "the effect", affect is an incorrect word here
  2. Results- each section describing the appropriate figures should be changed to summary statement, e.g. instead of saying: "Effect of Chymase Inhibitor on Neutrophil Infiltration into the Pancreas", the authors should state: "Chymase inhibitor TY-51469 decreases immune cell infiltration into the pancreas"
  3. Results- Fig.3.- did the authors measure serum lipase and pancreatic TNF-alpha at other time points? The effect of TY-51469 is not significant for serum lipase and seems minimal for pancreatic TNF-alpha?
  4. Results-Fig.4.-please provide better quality pictures
  5. How specific is the toluidine blue staining to clearly distinguish mast cells from other populations of immune cells?
  6. What is the effect of TY-51469 administration on other tissues? Please add into the discussion section
  7. Please add the most recent reviews/original articles discussing the effects of TY-51469 (e.g. Pejler G., J Innate Immun 2020)
  8. Methods: what is the vehicle for TY-51469, what is it dissolved in?
  9. Did the authors perform more stringent tests than LSD test?
  10. Please correct the manuscript for style- Fig.4.- please change "pancreases" to "pancreata"; it is inappropriate to use "complaining of" (Discussion section), should be "reporting", please change "inflammatory cytokines" to "proinflammatory cytokines"

Author Response

To Reviewer 1

Thank you for your valuable suggestions.

  1. We corrected “the affect” to “the effect”. (line 5)
  2. As recommended, we changed the summary statements as follows.

2.4. Chymase Inhibitor TY-51469 Decreases Immune Cell Infiltration into the Pancreas (line 118)

2.5. Chymase Inhibitor TY-51469 Suppresses Pancreatic Chymase and MMP-9 Activity (line 131)

2.6. Chymase Inhibitor TY-51469 Decreases the Number of Chymase-Positive Cells in the Pancreas (lines 141-142)

2.7. Chymase Inhibitor TY-51469 Increases the Survival Rate of Hamsters (line 155)

  1. We did not measure serum lipase and pancreatic TNF-α levels at other time points. At our university, there was a strict review of the 3Rs (Replacement, Reduction, Refinement) and it was pointed out that we should focus on only one time point to assess the effect of inhibitors. The reason why we chose 8 hours was that definite increases in blood markers, tissue inflammatory markers and inflammatory infiltration into the pancreas were observed at this time point.
  2. As recommended, we have presented clearer images in Figure 4.
  3. Mast cell granules can naturally induce metachromatic staining, which is important in the detection of mast cells and is strongly recommended as a routine stain for this purpose. The pictures in Figure 6 were not clear in the previous version, and we have presented better photographs in the revised version.
  4. We did not evaluate the effects of chymase inhibitors in tissues other than the pancreas. In previous reports, chymase inhibitors were found to exert anti-inflammatory effects in acute hepatitis and acute small intestinal ulcers. Acute pancreatitis leads to inflammation not only in the pancreas, but also in other organs such as the lungs, and it is possible that chymase inhibitors suppress inflammation in these other organs as well. We have added this description in the discussion section, as follows:

On the other hand, acute pancreatitis causes inflammation not only in the pancreas, but also in other organs such as the lungs, and it is possible that the chymase inhibitors suppressed inflammation in the other organs as well. In fact, chymase inhibitors, including TY-51469, have been shown to attenuate several inflammatory diseases other than pancreatitis [30]. (lines 257-261)

  1. Pejler, G. Novel insight into the in vivo function of mast cell chymase: lessons from knockouts and inhibitors. J Innate Immun 2020, 12, 357-372, doi: 10.1159/000506985. (lines 427-428)
  2. We have cited articles discussing the effect of TY-51469 in the revised version as above.
  3. We dissolved TY-51469 in saline in this study, and have described this in the methods section.

To evaluate the effect of TY-51469 on acute pancreatitis, 30 mg/kg of TY-51469 dissolved in saline or placebo (saline) was administered 1 h after the second injection of L-arginine, and blood and pancreatic samples were evaluated 8 h after the injection of L-arginine. (lines 274-277)

  1. We did not perform tests other than the LSD test.
  2. We corrected “pancreases” to “pancreata” (line 149), “complaining” to “reporting” (line 188), and “inflammatory cytokines” to “proinflammatory cytokines” (line 246).

Reviewer 2 Report

The study by Kuramoto et al demonstrated the chymase as a novel therapeutic target in Acute pancreatitis. The manuscript is well written, and results described in very simple tone. 

A few minor points are worth considering

  1. The survival rate was followed up to 24 hours, does the biomarker level remain same? what about IL-1β ? 
  2. I feel describing methods in brief though they performed using kit would be great. Missing survival evaluation methodology in method section.  
  3. I would strongly suggest citing figure numbers in discussion part. 
  4. Reference for line 229

Author Response

To Reviewer 2

Thank you for your valuable suggestions.

  1. We did not measure other biomarkers, including IL-1β.
  2. We did not measure the biomarkers and did not use kits for biomarkers. We have added details about survival evaluation methods in the revised method section.

To determine the effect of TY-51469 on the survival rate of L-arginine-induced pancreatitis, 30 mg/kg TY-51469 or placebo was administered 1 h after the second injection of L-arginine, and the resultant survival rates were calculated by dividing the number of hamsters that survived every 4 hours after L-arginine administration by the number of hamsters before the start of the study. (Lines 280-282)

  1. We added the figure number in the discussion section.

In the present study, serum lipase levels were significantly increased 1 h after the injection of 3.0 g/kg of L-arginine, and they were further increased at 2 h, as shown in Figure 2. (line 195)

In the present study, pancreatic MMP-9 activity was increased after L-arginine injection, but its activity was attenuated by treatment with TY-51469, although the difference was not significant, as shown in Figure 5. (line 216)

Furthermore, the significant increase in neutrophil infiltration of the pancreas observed in placebo-treated acute pancreatitis model hamsters was significantly ameliorated by treatment with TY-51469, as shown in Figure 4. (line 219)

Survival rates after the injection of L-arginine were significantly improved by treatment with TY-51469, as shown in Figure 7. (line 242)

  1. A reference was added on line 232 (line 229 in the previous version).

Several enzymatic functions of chymase have been demonstrated, and chymase might play a crucial role in increasing mast cell numbers [26].

Reviewer 3 Report

In the current manuscript titled “Chymase as a Novel Therapeutic Target in Acute Pancreatitis” Kuramoto et.al., have demonstrated effect of a chymase inhibitor in acute pancreatitis, authors have used L-arginine to induce acute pancreatitis in hamster model. Authors have investigated chymase inhibitor TY-51469. Overall, the hypothesis is not very well formed as they have not provided the rationale for using chymase as a target in acute pancreatitis and how it is related to the clinical setting. The manuscript is well written and important for the pancreatitis research; however, I have the following concerns.
Major concerns:
1. It would be of significant importance if authors have validated the chymase upregulation in normal human pancreas and human pancreatitis tissues.
2. The authors have demonstrated the change in the chymase activity upon induction of pancreatitis and decrease following the treatment of TY-51469. The authors also have to demonstrate the upregulation of chymase using IHC in normal murine pancreas and arginine treated pancreas.
3. Plasma amylase measurement is the most important hallmark of pancreatitis. Authors have to do the amylase measurement for dose dependent arginine treatment for pancreatitis induction as well as for the TY-51469 treated samples.
4. plasma concentrations of IL-1β and IL-18 will further strengthen the efficacy of the TY-51469.
5. Pancreas weight/body weight ratio has to be presented
6. The images are hard to see and high-quality images has to be presented so that it will be easy for the readers to see the images and understand the data.
Minor comments
1. Labeling on the figure would be easy for readers to understand. Such as place or treated group. Similarly, the IHC staining that was done.

Author Response

To Reviewer 3

Thank you for your valuable suggestions.

Major comments

  1. To the best of our knowledge, no previous paper has reported that chymase is upregulated in the pancreas of patients with acute pancreatitis.
  2. As shown in Figure 6, the number of pancreatic chymase-positive cells tended to increase in placebo-treated L-arginine hamsters compared with control (normal) hamsters. However, unlike the enzyme activity, the differences in chymase-positive cell numbers between normal and L-arginine-treated mice was not significant. We believe that the discrepancy between enzyme activity and histological analysis was due to the difference in analysis methods, although the trend was almost the same.
  3. As the reviewer pointed out, plasma amylase is an important marker in acute pancreatitis, and we agree with the referee's comments. We had the animals’ blood samples evaluated by SRL Co. Ltd. to measure amylase as well as lipase levels. However, although lipase levels could be measured in the hamsters, amylase levels could not. Therefore, we have only presented lipase levels.
  4. We used hamster as an animal pancreatitis model, and there was no kits for

measuring IL-1β and IL-18 in hamsters. Therefore, we did not measure plasma concentration of IL-1β and IL-18.

  1. Unfortunately, we forgot to assess pancreatic weight/body weight ratio in the present study.
  2. We changed the images in Figures 4 and 6, replacing them with more clear images.

Minor comments

  1. The labels in Figures 3-6 were changed.

Round 2

Reviewer 1 Report

The authors addressed the comments.

Reviewer 3 Report

The current hypothesis will be of great interest to readers in the field and very  important as this is very clinically relevant. The authors have addressed some of the comments, but however due to lack of availability or reagents and partly due to improper design they will not be able to execute other validations. Therefore I would recommend to accept in the present form as it cannot be further improved.